# CD36—A Host Receptor Necessary for Malaria Parasites to Establish and Maintain Infection

**DOI:** 10.3390/microorganisms10122356

**Published:** 2022-11-29

**Authors:** Anna Bachmann, Nahla Galal Metwally, Johannes Allweier, Jakob Cronshagen, Maria del Pilar Martinez Tauler, Agnes Murk, Lisa Katharina Roth, Hanifeh Torabi, Yifan Wu, Thomas Gutsmann, Iris Bruchhaus

**Affiliations:** 1Bernhard Nocht Institute for Tropical Medicine, 20359 Hamburg, Germany; 2Biology Department, University of Hamburg, 22601 Hamburg, Germany; 3Centre for Structural Systems Biology, 22607 Hamburg, Germany; 4German Center for Infection Research (DZIF), Partner Site Hamburg-Borstel-Lübeck-Riems, 20359 Hamburg, Germany; 5Division of Biophysics, Research Center Borstel-Leibniz Lung Center, 23845 Borstel, Germany

**Keywords:** *Plasmodium falciparum*, malaria, sequestration, cytoadhesion, endothelial cell receptor, CD36

## Abstract

*Plasmodium falciparum-*infected erythrocytes (*Pf*IEs) present *P. falciparum* erythrocyte membrane protein 1 proteins (*Pf*EMP1s) on the cell surface, via which they cytoadhere to various endothelial cell receptors (ECRs) on the walls of human blood vessels. This prevents the parasite from passing through the spleen, which would lead to its elimination. Each *P. falciparum* isolate has about 60 different *Pf*EMP1s acting as ligands, and at least 24 ECRs have been identified as interaction partners. Interestingly, in every parasite genome sequenced to date, at least 75% of the encoded *Pf*EMP1s have a binding domain for the scavenger receptor CD36 widely distributed on host endothelial cells and many other cell types. Here, we discuss why the interaction between *Pf*IEs and CD36 is optimal to maintain a finely regulated equilibrium that allows the parasite to multiply and spread while causing minimal harm to the host in most infections.

## 1. Introduction

Despite progress in malaria control, malaria remains one of the most important infectious diseases worldwide. In 2020, about 267 million malaria cases, including 409,000 deaths, were recorded [1]. Concerning all malaria parasites, the deadliest, *Plasmodium falciparum*, has a complex life cycle that alternates between *Anopheles* mosquitoes and humans. The asexual cycle that takes place in humans consists of the liver stage (multiplication of the parasite in hepatocytes) and the intraerythrocytic cycle (multiplication of the parasite in erythrocytes). Each intraerythrocytic cycle lasts approximately 48 h, during which the merozoite that invaded the erythrocyte develops through the ring stage to the trophozoite and finally the schizont. At the end of the intraerythrocytic cycle, the newly formed merozoites are released, ready to invade new erythrocytes. Some merozoites develop into gametocytes, which must be taken up by female Anopheles mosquitoes to complete their sexual development. The complexity of the parasite’s life cycle and its masterful ability to evade its elimination by the host immune system challenge our efforts to combat the disease.

To survive in the human host, *P. falciparum* has evolved unique mechanisms, two of which, called sequestration and antigenic variation, rely mainly on a highly diverse protein family, the *P. falciparum* erythrocyte membrane protein 1 (*Pf*EMP1). The *Pf*EMP1s, which the parasite exposes on the surface of its host cell from the trophozoite stage onwards, have at least a dual function. First, they bind to various endothelial cell receptors (ECRs) on the walls of blood vessels (sequestration), thereby disappearing from the peripheral circulation and bypassing removal by the spleen. Unlike other plasmodial species, the deformability of *P. falciparum* infected erythrocytes (*Pf*IEs) decreases as the parasite matures, so that circulating trophozoites and schizonts would be retained in the spleen and removed from circulation by resident macrophages [2,3,4,5,6]. Second, *Pf*EMP1 represents the main target of the humoral immune response [7], but due to the presence of numerous copies of *var* genes encoding *Pf*EMP1, the parasite can sequentially present different *Pf*EMP1 variants on the surface of its host cell and use them for sequestration. The ability to alter the presented *Pf*EMP1 by antigenic variation enables the parasite to stay one step ahead of the immune system and maintain long-lasting, chronic infections, e.g., for bridging dry seasons [8,9,10,11,12]. 

## 2. The *Pf*EMP1 Family

The *Pf*EMP1 family is encoded by about 45–90 *var* genes per parasite genome [12]. Expression of the *var* genes is mutually exclusive in ring-stage parasites, such that only a single *Pf*EMP1 variant is present on the surface of trophozoite- or schizont-stage *Pf*IEs at any given time [13,14] for review [10]. Mutually exclusive expression relies on very complex mechanisms. These are based on both epigenetic regulation and cis-acting DNA elements and RNA transcripts involved in *var* gene activation and silencing (for review [10]). 

The *var* genes and their encoding *Pf*EMP1s vary greatly from parasite to parasite, and recombination constantly generates new variants, so there is an enormous repertoire of *var* genes in nature [13,14,15,16]. The molecular masses of *Pf*EMP1s range from 150 to 400 kDa. These proteins consist of an intracellular acidic terminal segment (ATS domain), a transmembrane domain, and a variable, extracellularly exposed region responsible for receptor binding. This extracellular region contains a single N-terminal segment (NTS; main classes A, B, and pam) and a variable number of different Duffy binding-like domains (DBL; main classes α–ζ and pam) and cysteine-rich interdomain regions (CIDR; main classes α–δ and pam) [17,18,19,20]. Approximately two-thirds of *var* genes localize in the subtelomeric regions of the chromosomes. Most of the subtelomeric and central localized *var* genes are located in regions of electron-dense heterochromatin at the nuclear periphery, with the active *var* gene shifting to the region of lower electron density [14]. Depending on the chromosomal localization, the upstream sequence, and the direction of transcription of the *var* genes, *Pf*EMP1s can be classified as A, B, C, or E [17,21,22,23].

A few conserved, strain-transcendent *var* variants have been described: *var1*, *var2csa* (group E), and *var3*. The *var1* gene occurs in two variants in the parasite population, *var1-3D7* and -*IT*, is often conserved as a pseudogene, and the encoded protein may not be presented on the erythrocyte surface [12,24]. VAR2CSA has an atypical domain architecture, mediates binding to chondroitin sulfate A (CSA) in the placenta, and is, thus, important in pregnancy-associated malaria [25]. VAR3 proteins are very short *Pf*EMP1s with unknown receptor binding phenotypes [26]. Analysis of 399 different *Pf*EMP1 sequences from seven *P. falciparum* genomes allowed the identification of 23 domain cassettes (DCs) that could be important for protein folding and binding to human ECRs, as well as for reflecting recombination breakpoints [17]. About 10% of *Pf*EMP1s variants belong to group A and are usually longer proteins with a head structure that includes a DBLα1 and either a CIDRα1 domain (CIDRα1.4–7) that binds to the endothelial protein C receptor (EPCR) or a CIDRβ/γ/δ domain with unknown receptor binding phenotype [12]. Groups B and C make up the majority of *Pf*EMP1s (at least 75%) and typically have DBLα0-CIDRα2–6 head structures that bind to CD36, followed by only two additional extracellular domains (DBLδ1, CIDRβ/γ). A subset of chimeric B-type proteins (group B/A, also known as DC8-containing proteins) has a DBLα2 domain (chimeric DBLα0/1 domain) and an EPCR-binding CIDRα1.1 or CIDRα1.8 domain typically attached to a complement component C1q receptor (C1qR)-binding DBLβ12 domain [27,28,29,30,31,32,33,34]. Thus, the head structure confers mutually exclusive binding properties to either EPCR (14%), CD36 (72%), CSA (3%), or to one or more unknown ECRs via the CIDRβ/γ/δ domains (10%) or VAR3 (1%) [35]. Concerning the C-terminal to the head structure, most *Pf*EMP1s have additional DBL domains, of which certain subsets of the DBLβ domains bind intercellular adhesion molecule-1 (ICAM-1) [36,37] or C1qR [38]. As an example, the *Pf*EMP1 repertoire of the *P. falciparum* isolate IT4 is shown in Figure 1. 

## 3. Knobs—Anchor Point for *Pf*EMP1s

*Pf*EMP1s are concentrated in nanoscale, electron-dense protrusions of the plasma membrane of *Pf*IEs, the so-called knobs. They are formed in erythrocytes about 16 h after parasite invasion and reach their highest density 20 h after infection [39,40]. Single knobs have a hemispherical ellipsoid shape with a minor axis of 20 nm and a major axis of 120 nm [41]. Knobs are composed of various submembrane structural proteins, including the major protein of this structure, knob-associated histidine-rich protein (KAHRP). These consist of *Pf*EMP3, the ring-infected red cell antigen (RESA), the mature parasite-infected red cell surface antigen (MESA)/*Pf*EMP2, and *Pf*332 [41,42]. The knobs consist of a highly organized skeleton made of a spiral structure located beneath specialized areas of the erythrocyte membrane (Figure 2) [43]. The arrangement of *Pf*EMP1s in a cluster near the top of the knobs is assumed to increase the binding capacity of *Pf*IEs, especially under flow conditions (see below) [44,45,46,47].

## 4. Endothelial Cell Receptors (ECRs)

At least 24 ECRs were described as binding partners for *Pf*IEs. These include EPCR, gC1qR, ICAM-1, and CD36, mentioned above, as well as platelet endothelial cell adhesion molecule-1 (PECAM-1), CSA (adhesion to placental epithelium) [49], heparan sulphate, hyaluronic acid, neuronal cell adhesion molecule (NCAM), P-selectin, E-selectin, vascular cell adhesion molecule-1 (VCAM-1), thrombospondin, fractalkine, ανβ3- and αVβ6-integrin, fibronectin, CD9, CD151, multidrug resistance protein 1 and 2, erythropoietin receptor 1, and tumour necrosis factor receptor (TNFR) 1 and 2 [6,33,37,48,50,51,52,53]. To date, only a few ECRs have been shown to interact via *Pf*EMP1, and *Pf*EMP1 binding domains have only been identified for CD36, ICAM-1, EPCR, PECAM-1, and gC1qR [18,27,32,33,54]. 

## 5. Cytoadhesion of *Pf*IEs

Cytoadhesion of *Pf*IEs to ECRs in the vascular bed of organs, such as the brain, heart, lung, stomach, skin, and kidney is a central component of the pathogenesis of malaria [3,4,5,6,55,56,57]. In addition to the blockage of capillaries by the cytoadhesion of the *Pf*IEs, there is also an increased production of inflammatory cytokines, endothelial dysfunction, and increased vascular permeability in the affected tissue [58,59]. As a result of the immune response triggered by the growth and sequestration of the parasites, patients develop fever, headache, muscle pain, and rigor [3,60,61,62,63]. Depending on the age and immune status of the patient, severe, fatal complications such as cerebral malaria (CM), lung damage, kidney failure, acidosis, and severe anemia may occur [3,62,63]. Both children and adults can be affected by cerebral malaria, but while severe malaria puts children at higher risk for anemia and convulsions, liver dysfunction and kidney failure are more common in adults. In addition, the clinical picture of severe malaria clearly depends on age, with mortality increasing significantly with age [64].

## 6. Pathology Induced by Cytoadhesion

Different *Pf*EMP1s have different binding properties to ECRs and are associated with different clinical outcomes (see review [4]). Several studies have shown that severe malaria is associated with the expression of group A and B/A *Pf*EMP1s and, in particular, with variants possessing EPCR binding capacities [65,66,67,68,69]. In contrast, infections dominated by CD36-binding parasites show mild disease courses [33,35,68,70,71,72].

Since *Pf*EMP1s are multi-domain proteins, it has already been shown that some variants can mediate adhesion to multiple ECRs (dual binder) [36,37,73,74]. Examples include *Pf*EMP1 variants that interact with ICAM-1 and EPCR or CD36. Dual binding to ICAM-1 and EPCR specifically enhances the binding of *Pf*IEs to endothelial cells (ECs) under physiologically higher shear stresses. Expression of these variants has been associated with an increased risk of developing CM, including induction of brain swelling and disruption of the blood-brain barrier [36,74,75,76]. Less is known about the role of dual ICAM-1 and CD36 binding *Pf*EMP1s, mainly of group B, but ICAM-1 and CD36 have been shown to work together to enhance the binding of *Pf*IEs to microvascular cells [27,37,66,77,78].

## 7. ECR-Specific Expression in Relation to the Origin of the Endothelial Cells

ECs derived from different organs presenting different ECRs on their cell surface. It is known that EPCR and ICAM-1 are presented on brain ECs and are mainly bound by DC8- and group A *Pf*EMP1s [33,36,74,75,79]. Ortolan and colleagues have recently shown that the same *Pf*EMP1s that cytoadhere to brain ECs also bind EPCR intestinal and renal ECs. In this context, it is suggested that a binding axis between the brain, gut, and kidney may contribute to the multi-organ complications of severe malaria [80]. In contrast, CD36 is not presented on brain ECs, or only in very low amounts, and was also not detected on intestinal and peritubular renal ECs [81,82]. Thus, in contrast to EPCR, CD36 seems to occur mainly in the microvascular beds of non-vital organs. As mentioned above, several studies have linked EPCR- or dual EPCR- and ICAM-1-binding *Pf*EMP1s to severe malaria, which most likely occurs in individuals without preformed immunity [33,68,72,79,83,84,85]. For example, Wichers and colleagues recently demonstrated a clear association between the expression of *Pf*EMP1s, which have an EPCR-binding phenotype, with first-time infection and severe malaria [84]. Transcripts for CD36-binding variants were found more frequently in parasites from non-severely infected and pre-exposed patients. 

Interestingly, however, in the same study, CD36-binding variants are overrepresented in all groups of adult malaria patients analyzed, even in severe cases and in first-time infected individuals [84], which is in stark contrast to the pattern seen in severely ill children [70]. The authors speculate that this could be the reason for multisystemic disease symptoms in adult malaria patients. Alternatively, parasites in these less ill adult patients compared to children could have a less dominant expression of EPCR-binding *Pf*EMP1 [84]. Further studies also showed that parasite cytoadhesion to CD36 correlates with the development of mild malaria [70,85,86]. Accordingly, both factors, the already acquired immunity and the age of patients, seem to favor the expression of CD36-binding variants.

## 8. Hierarchy of *var* Expression during the Human Blood Phase

Independent analyses of first-generation blood-stage parasites from malaria-naive human volunteers infected with *P. falciparum* sporozoites have shown remarkably consistent expression of a broad repertoire of *var* genes, primarily type B (*P. falciparum* strain NF54: [87,88,89,90]; unpublished data for *P. falciparum strain* 7G8). This broad expression pattern is modulated by existing host immunity. In African pre-exposed individuals, the expression of many variants at the parasite population level is reduced to very few or a single B-type, possibly reflecting gaps in the host antibody repertoire [91]. In severe disease, expression of *var* genes shifts toward group A or A/B for unknown reasons, resulting in expression of *Pf*EMP1 with EPCR and/or ICAM-1 or a yet unknown binding domain [65,84,92,93,94,95,96]. Group A *Pf*EMP1s are therefore thought to possess binding phenotypes that confer a selective advantage for parasites to replicate asexually, e.g., by decreasing splenic clearance, but at the same time favors the development of severe malaria [31]. In this context, it has been shown that antibodies for the EPCR binding domains (CIDRα1.1/4–8) are acquired faster and earlier in life than those to the CD36 binding domains (CIDRα2–6) in endemic areas and that this is associated with protection against severe malaria, including CM [31,65,95,97,98]. This raises the question of the evolutionary advantage of A-type expression for the parasite since cytoadhesion in vital organs via EPCR and ICAM-1 may lead to rapid death of infected individuals and thus not to transmission of the parasite to the mosquito. On the other hand, the rapid development of the protection of individuals from severe malaria could therefore be an advantage for the parasites, as they would be less likely to harm their hosts in the event of re-infection [93]. Later in the course of asymptomatic infection, *Pf*IEs appear to have altered cytoadherence properties, as more developed, “older” parasites circulate in the blood than in symptomatic cases. This observation suggests that these parasites either express a lower total amount of *Pf*EMP1 on the host cell surface or a different set of *Pf*EMP1 variants with less adhesive binding domains [99]. Since it is already known that chromosomal location determines the on-and-off rate of *var* genes [100], it would be plausible that parasites in primary infections initiate expression of the most telomeric B-type *var* genes, then tend to express A-types during severe disease, but in the case of long-lasting asymptomatic infections may then express centrally located C-types, which are known to have very slow off rates in comparison to subtelomeric *var* genes. The concept of an initial high *var* gene switching rate to establish infection and a slower switching rate of later expressed genes to maintain infection was already proposed 20 years ago [101,102].

## 9. *P. falciparum* and CD36

Looking at the *Pf*EMP1 family, the question arises why, depending on the parasite genome, between 75–85% of *var* genes encode *Pf*EMP1s, which have a CIDRα2–6 domain for CD36 binding [17,27,31]. Interestingly, the CIDRα domains were shown to be present only in the *P. falciparum*-containing branch (clade B) of the *Laverania* subgenus. This could indicate that the binding to CD36 provides a selective advantage for *P. falciparum* [103]. What kind of selection advantage this was is yet unclear.

What advantage does the parasite have in retaining this large number of CD36-binding *Pf*EMP1 variants in its genome? Additionally, what is the difference between the individual variants or, more generally, between CD36 binding mediated by group B or C *Pf*EMP1s? 

## 10. CD36

CD36 is a pattern recognition receptor (PRR) that belongs to the class B scavenger receptor family. It is a glycoprotein present in many tissues and involved in several key processes. These include lipid processing and uptake, thrombostasis, glucose metabolism, immune function, angiogenesis, and fat taste (for review [104,105,106,107,108]. CD36 is found on platelets, mononuclear phagocytes, adipocytes, hepatocytes, myocytes, some epithelia and, as mentioned above, expressed on the endothelia of liver, spleen, skin, lung, muscle, and adipose tissue [81,82,109]. On microvascular ECs, CD36 is a receptor for thrombospondin-1 and related proteins and functions as a negative regulator of angiogenesis. At least 60 variants have been described in the coding region of the *CD36* gene. The mutations of CD36 caused by gene variants can also influence the adhesion of *Pf*IEs and ECs. This could directly influence the severity of a malaria infection via the degree of cytoadhesion. There are several studies on this, but with contradictory results [110]).

## 11. CD36 Binding *Pf*EMP1 Variants—Benefits for Parasite and Host

Several observations may help explain why a large number of CD36-binding *Pf*EMP1 variants is not only beneficial for parasite development, but may also be an advantage for the infected host. 

The parasite targets a region of CD36 that is essential for its physiological role in fatty acid uptake because mutation of F153 disrupts the interaction of CD36 with CIDRα2–6 but also abolishes the binding of CD36 to oxidized LDL particles. This reduces the likelihood that the human host can escape from *Pf*EMP1 binding by altering its CD36 [28].In contrast to the EPCR binding surface of CIDRα1 domains, which protrudes and is a structure that is likely to be well recognized by antibodies, the CD36 binding site is concave, and the conserved hydrophobic residues are hidden in a pocket, so maybe they are less easily recognized. In addition, the binding site is surrounded by a sequence-diverse protein surface containing a flexible loop that may make antibody recognition less likely. This unique interaction site of the parasite with CD36, which protects essential residues from exposure to the immune system, appears to allow the parasite to utilize an antigenically diverse set of CIDRα2–6 for cytoadhesion to CD36 to be protected from splenic clearance [28].CD36 is found in cells of the innate and adaptive immune system [104,105,106,107,108]. It has been shown that *Pf*IEs can adhere to dendritic cells (DCs). This attachment inhibits maturation of these cells and their ability to stimulate T cells. Thus, the parasite can trigger dysregulation of the immune system. This favors the development of the parasite by impairing the host immune system’s ability to clear the infection [108,111,112,113,114]. However, there is also an observation that the mechanism of DC inhibition by *Pf*IEs may be independent of *Pf*EMP1 and CD36 [115].The previously determined hierarchy of *var* expression upon parasite entry into human blood begins with group B and suggests that most parasites bind to CD36, as they all encode a CD36-binding phenotype. Most infected individuals, including those who are not immune, do not develop severe malaria, and cytoadhesion of *Pf*IEs occurs in extensive microvascular beds in tissues other than the brain (skin, muscle, adipose tissue). Therefore, cytoadhesion in such non-vital tissues could promote survival and transmission of the parasite while minimizing host damage and death [87,88,89,90].Antibody-induced selective binding and internalization of CD36 do not result in proinflammatory cytokine production by human macrophages. Interestingly, CD36-mediated phagocytosis of *Pf*IEs also did not result in cytokine secretion by primary macrophages [116]. However, CD36-mediated binding of *Pf*IEs increases the likelihood of phagocytosis by macrophages. This can lead to a reduction in parasitemia, but also allows the parasite to maintain a viable infection without causing too much damage to the host through high parasitemia [108,114,117,118].DCs react to *P. falciparum* very early during infection and can, thus, influence the development of immunity. Internalization of *Pf*IEs by DCs and subsequent pro-inflammatory cytokine production of DCs, NK, and T cells depends on CD36. Notably, plasmacytoid DCs regulate innate and adaptive immunity to malaria via the production of proinflammatory cytokines. As this effect is particularly evident at low levels of parasitemia, the role of CD36 for malaria immunity appears to take place early during infection and to promote the development of protective immunity against malaria [118,119].

All these observations underline the importance of CD36 for malaria. During long co-evolution, a fine balance has evolved between host and parasite, allowing the parasite to multiply but harming the host as little as possible.

## 12. Binding Phenotypes of *Pf*IEs

Cytoadhesion of *Pf*IEs is divided into the three phases: “tethering”, “rolling”, and “immobilization”, comparable to leukocyte diapedesis [120,121,122]. However, the dynamics of cytoadhesion of *Pf*IEs to the vascular endothelium is controversial. For example, some authors describe cytoadhesion to ICAM-1 as rolling, and to CD36 as stationary, or vice versa [123,124,125,126,127]. However, there is increasing evidence that *Pf*IEs are very likely to roll over CD36 [126,127,128,129,130,131]. Recently, the binding phenotype for different ECRs was investigated using a laminar flow system with transgenic Chinese hamster ovary (CHO) cells carrying different ECRs on their surface [127]. Rolling was observed upon interaction with CD36, and the rolling behavior of disc-shaped *Pf*IEs at the trophozoite stage (flipping) differed from the rolling behavior of round-shaped *Pf*IEs at the schizont stage (continuous rolling) (Figure 3). Moreover, *Pf*IEs in the schizont stage roll more stably than *Pf*IEs in the trophozoite stage at different shear stresses [127]. The rolling motion of *Pf*IEs was also seen on transgenic mouse fibroblasts presenting CD36 [128] and on recombinant CD36 instead of transgenic eukaryotic cells [129]. As described above, the dermal endothelium has large amounts of CD36. Rolling movements of *PfI*Es have also been found on dermal ECs, as well as on human skin grafts, on which large amounts of CD36 are found [126,128,130,131]. Additionally, last but not least, the rolling CD36 binding phenotype was also confirmed by in silico modeling [132,133]. However, depending on the experimental setup, the parasite isolates used, and the parasite stage, different velocities were measured at similar shear forces. For trophozoite-stage parasites confronted with recombinant CD36, average velocities between 140 µm/min to 680 µm/min were measured at a shear force of 1.6 Pa, depending on the isolate [129]. When transgenic CHO cells presenting CD36 on the surface were used instead of recombinant CD36 in a similar experimental setup, average velocities ranging from 11 µm/min to 33 µm/min, i.e., a 12–20 fold lower value, were measured, also depending on the parasite stage and isolate [127]. If *Pf*IEs cytoadhere for approximately 30 h during their intraerythrocytic development, they travel distances between 25–122 cm or 2–6 cm, respectively, depending on the experimental setup [127,129]. In both cases, however, the probability of passing over the spleen and being removed accordingly is low.

Further studies showed that initial contact of *Pf*IEs to CD36 under flow conditions activates Scr-family kinases, leading to dephosphorylation of CD36 via p130CAS signaling. This increases the binding affinity of *Pf*IEs to CD36 and, thus, leads to increased adhesion of the *Pf*IEs. This mechanism also leads to actin cytoskeletal remodeling and subsequent CD36 clustering, which further increases *Pf*IE adhesion [128,131,134]. It is postulated that a small number of strongly adherent *Pf*IEs activate the endothelium, and thus enhance the cytoadhesion of most parasites [131]. However, the binding mode of *Pf*IEs also seems to be strongly dependent on the respective ECR. For ICAM-1, CD9, P-selectin, as well as CSA, stationary binding, instead of rolling, was observed under flow conditions [127] (Figure 3). Stationary binding to ICAM-1 was also demonstrated in an earlier study [126]. However, while binding to CD36 occurred at shear forces below 4 dyn/cm^2^, binding to ICAM-1, CD9, P-selectin, and CSA occurred mostly at lower shear forces (from 2 dyn/cm^2^) [127].

Of note, the origin and environment of the ECR studied (recombinant or presented on eukaryotic cells) also seems to be important for characterising the binding phenotype. Antia and colleagues observed a rolling binding type for *Pf*IEs, with an average rolling velocity of about 10 µm/s at 1–2 kPa and of 1–3 µm/s when recombinant ICAM-1 or CD36 was used, respectively [123]. Interestingly, in the same study, stationary binding of *Pf*IEs, as also described by Lubiana and colleagues [127], was observed on transgenic CHO cells presenting ICAM-1 [123]. However, the binding showed large variations. Thus, the *Pf*IEs came to a standstill for a few seconds, but were then also able to detach from the CHO cells again [123].

## 13. Importance of Knobs for Cytoadhesion

It is well established that knobs play a crucial role in the cytoadhesion of *P. falciparum* [45,47,48,127,135]. Among other findings, *Pf*IEs from patients with hemoglobin S (HbS) or hemoglobin C (HbC; both hemoglobin mutations protect against severe complications and death from malaria [136,137,138]) have been found to exhibit reduced cytoadhesion to microvascular endothelial cells [139,140]. The results suggest that HbS and HbC alter the erythrocyte membrane in a manner that inhibits the transport and/or docking of parasite proteins and impairs the ability of the parasite to remodel the surface of its host cell. This also leads to the fact that the knobs can no longer be formed correctly, and *Pf*EMP1 is also no longer presented correctly [139,140].

In the absence of knobs, parasite adhesion to CD36 was observed only under static conditions, but not under flow conditions simulating the situation in human blood [47,141]. In the absence of knobs on the surface of *Pf*IEs, the rolling distance is shortened compared to knob-positive *Pf*IEs [127]. The more stable rolling described above, and the rolling over longer distances of the schizont stage, is most likely related to the uniform coverage of the surface with knobs [130]. Furthermore, the adhesion force seems to be lower in the schizonts than in the trophozoites [142]. The comparison of knob-negative and knob-positive *Pf*IEs suggests that the presence of knobs stabilises the ligand-receptor interaction due to the concentrated amount of *Pf*EMP1 on the knob surface (Figure 4) [46,127].

Another important observation could be made under fever conditions. Only knob-positive *Pf*IEs were able to bind to different ECRs (CSA, CD36) at 40 °C with preserved ECR-specific binding mode (rolling or stationary) (Figure 4) [48,127]. Measurement of the binding force between *Pf*IEs and CSA by force spectroscopy showed a decrease in binding force at febrile temperatures, but the number of bound *Pf*IEs increased. It was hypothesized that this increase in binding is due to non-specific binding despite the decrease in force [143]. Again, however, a study showed that, at febrile temperatures, binding affinities to CD36 and ICAM-1 decreased [144]. 

In summary, there is strong evidence that the presence of knobs on the surface of *Pf*IEs is an essential prerequisite for the parasite circulating in the bloodstream to adhere to the endothelium even under febrile conditions. An evolutionary pressure for the formation of knobs on *Pf*IEs in the human host is therefore operative.

## 14. Conclusions

CD36 is the main receptor for *Pf*IE cytoadhesion to the vascular endothelium. Due to the rolling behavior and the resulting short contact of parasites with CD36 on the ECs, these ECs may not or are only slightly activated. Likely, only B group *Pf*EMP1s with a particularly strong binding affinity to CD36 or dual binding properties, as well as the increase in parasitemia and the accompanying stimulation of the immune system and the release of proinflammatory cytokines, lead to an activation of the endothelium and thus also to the presentation of other ECRs such as ICAM-1 or P-selectin. Finally, *Pf*IEs with different binding phenotypes can also adhere, with static binding leading to further activation of the endothelium and the immune system.

Further observations highlight the role of CD36 in *P. falciparum* infection. (i) A large number of *Pf*EMP1s containing a CD36 binding domain [17,27,31]; (ii) the binding of *Pf*IEs to DCs via CD36, which inhibits their cell maturation and ability to stimulate NK and T cells [108,111,112,113,114]; (iii) CD36-mediated phagocytosis of *Pf*IEs does not result in cytokine secretion by macrophages [116] (however, it may result in a reduction in parasitemia [108,114,117,118]); (iv) in the early stage of infection, internalization of CD36-binding *Pf*IEs by DCs leads to increased cytokine production and activation of NK and T cells, which promotes the establishment of protective immunity [118,119].

Thus, CD36 is of great importance for establishing a finely regulated equilibrium between the parasite and the host, whereby the parasite can multiply and spread while the host experiences little damage. 

## Figures and Tables

**Figure 1 microorganisms-10-02356-f001:**
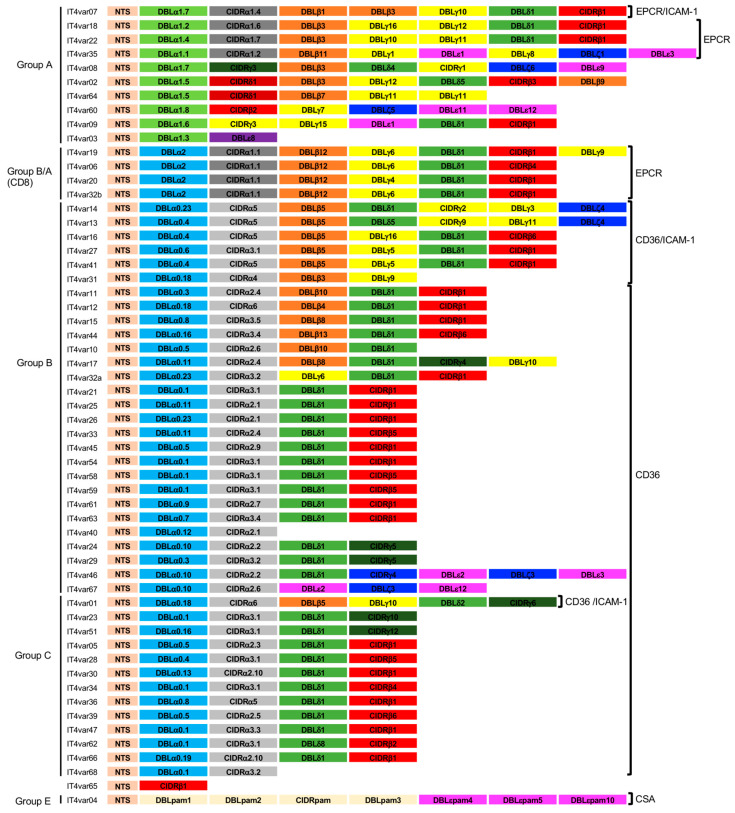
*Pf*EMP1 repertoire of the *P. falciparum* isolate IT4, adapted from [17]. ECR binding phenotype [36]. Color code: light brown: N-terminal segment (NTS); bright green: Duffy binding-like (DBL)α1; light blue: DBLα2, DBLα0; dark grey: Cys rich inter-domain regions (CIDR)α1; light grey: CIDRα2–6; dark green: CIDRγ; orange: DBLβ; yellow: DBLγ; green: DBLδ; pink: DBLε; blue: DBLζ; purple: DBLε; IT4var04: light yellow: DBL/CIDRpam: pink: DBLεpam.

**Figure 2 microorganisms-10-02356-f002:**
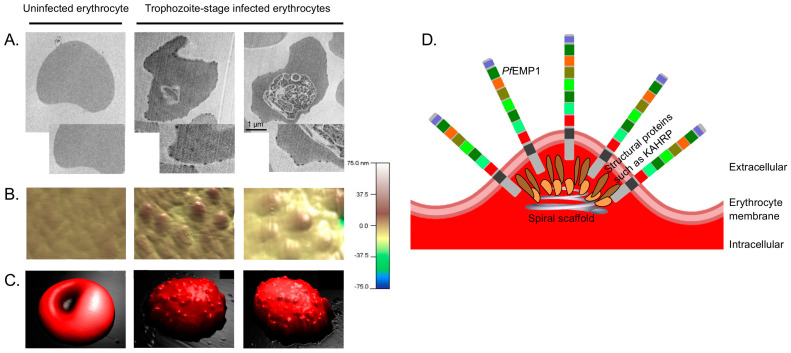
The knobs of *Pf*IEs. (**A**) Transmission electron micrographs of uninfected and synchronised trophozoite-stage *P. falciparum* culture 24–28 h post-infection. The parasites were cultivated in the presence of human serum (10%), and the *Pf*IEs were subjected to gelatin sedimentation to enrich knobby *Pf*IEs [40,48]. (**B**,**C**) atomic force microscopic three-dimensional images of the surface of uninfected and trophozoite-stage *Pf*IEs [40]. Images in (**B**) show magnifications directly from the membrane surface of the erythrocytes shown in (**C**). (**D**) Simplified schematic illustration of the structure of knobs.

**Figure 3 microorganisms-10-02356-f003:**
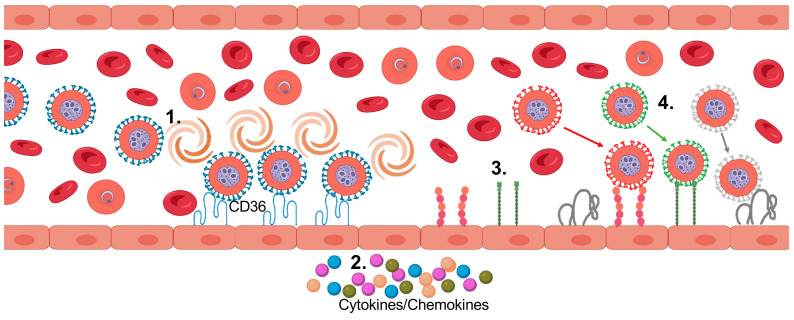
Presumed sequence of sequestration of *Pf*IEs to the vascular endothelium. 1. Adhesion and rolling over CD36. 2. Over time: endothelial activation, cytokine release. 3. Cytokine/chemokine-induced presentation of various receptors (e.g., ICAM-1, P-selectin, CD9). 4. Adhesion of *Pf*EMP1s with different binding phenotypes (created with BioRender).

**Figure 4 microorganisms-10-02356-f004:**
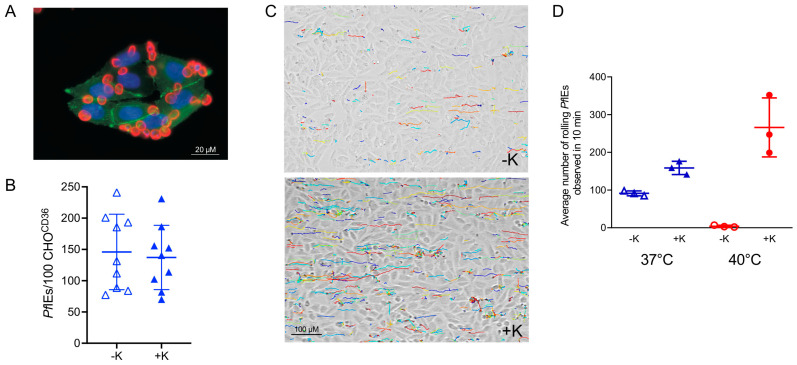
Cytoadhesion of knobby and knobless *Pf*IEs to transgenic CHO cells presenting CD36 on the cell surface (CHO^CD36^) under static and flow conditions and at different temperatures. (**A**) Adhesion of *Pf*IEs (red; anti-glycophorin A) to CHO^CD36^ cells (blue: nucleus (DAPI), green: cell surface (CD36-GFP fusion protein) under static binding conditions. (**B**) Cytoadhesion of knobbless (−K) and knobby (+K) *Pf*IEs to CHO^CD36^ cells. (**C**) Trajectories showing the rolling binding behavior of knobless (−K) and knobby (+K) *Pf*IEs to CHO^CD36^ cells. (**D**) Average number of knobless (−K) and knobby (+K) *Pf*IEs adhering to CHO^CD36^ cells at 37 °C (blue) and 40 °C (red) at a shear stress of 0.9 dyn/cm^2^ [127].

## Data Availability

Not applicable.

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
