# Peer review of "CD36—A Host Receptor Necessary for Malaria Parasites to Establish and Maintain Infection"

_microorganisms, 2022, doi:10.3390/microorganisms10122356_

Round 1

Reviewer 1 Report

See attachment.

Author Response

Response to Reviewer 1

We would like to thank the reviewer for the comments and suggestions for improvement. Below you will find our corresponding corrections.

  1. Line 51. This sentence currently refers to PfEMP1 and reads “due to the presence of numerous copies in the genome….”. Technically speaking, there are no copies of PfEMP1 in the genome, but rather copies of the encoding genes. Therefore, this would be better written “due to the presence of numerous copies of the genes encoding PfEMP1…..” or something similar.

Corrected – Line 59-62:

“Second, PfEMP1 represent the main target of the humoral immune response [7], but due to the presence of numerous copies of var genes encoding PfEMP1, the parasite can sequentially present different PfEMP1 variants on the surface of its host cell and use them for sequestration.”

  1. Line 67. Here the authors introduce the var gene family and state that var genes are mutually exclusively expressed. It would be good to mention that this is an epigenetic process and provide a citation or two.

Corrected – Line 67-72:

“The PfEMP1 family is encoded by about 45-90 var genes per parasite genome [12]. Expression of the var genes is mutually exclusive in ring-stage parasites, such that only a single PfEMP1 variant is present on the surface of trophozoite- or schizont-stage PfIEs at any given time [13,14] for review [10]. Mutually exclusive expression relies on very complex mechanisms. These are based on both epigenetic regulation and cis-acting DNA elements and RNA transcripts involved in var gene activation and silencing (for review [10]).” 

  1. The sentence beginning on line 77 describes the classification of PfEMP1 types and mentions chromosome location and direction of transcription, but provides no context. It would be best to first state that var genes are found in regions of the genome prone to the formation of heterochromatin, in particular the subtelomeric regions of the chromosomes. On line 79, the direction of transcription is with respect to the telomere.

Corrected – Line 81-87:

“Approximately two-thirds of var genes localize in the subtelomeric regions of the chromosomes. Most of the subtelomeric and central localized var genes are located in regions of electron-dense heterochromatin at the nuclear periphery, with the active var gene shifting to the region of lower electron density [14]. Depending on the chromosomal localization, the upstream sequence, and the direction of transcription of the var genes, PfEMP1s can be classified A, B, C, or E [17,21-23].”

  1. Lines 81 and 82. Shouldn’t var2csa and var3 also be italicized, similar to var1?

Corrected – Line 87-88:

“A few conserved, strain-transcendent var variants have been described: var1, var2csa (group E), and var3.”

  1. Line 110 begins with “This”, which seems out of place.

Corrected.

  1. Line 152. This sentence reads “In addition, mortality increases with age.” This sentence should be clarified somewhat since overall mortality due to malaria decreases with age. Perhaps something like “Amongst patients suffering from cerebral malaria, mortality increases with age.”

Corrected – Line 164-165:

“In addition, the clinical picture of severe malaria clearly depends on age, with mortality increasing significantly with age [64].”

  1. Lines 177-178. These sentences seem to indicate that the liver and lungs are not vital organs. This is in contrast to most definitions.

Line 188

The reviewer is right. We have deleted the sentence.

  1. In the paragraph beginning on line 186, the authors differentiate between adults and children with severe malaria. It is not clear from this description if the important characteristic is the increased immunity that adults in endemic regions have acquired through previous infections or if the important attribute is the actual age of the patient. If this is known, they authors should add this to the discussion.

As written, it does indeed seem confusing, so thank you for pointing out this inaccuracy. In the first part of the sentence, we have now cited the study by Wichers et al. in which the overrepresentation of CD36-binding variants was observed in adult malaria patients. These adults were returnees from Africa with first-time malaria infection or infrequent preexposure due to previous visits. Although there is a clear association between EPCR-binding variant expression and severe disease and naïve immune status, the frequency of CD36-binding variants is greatly increased in all adults examined in the study compared with the frequency observed in children by Jespersen et al. Accordingly, it appears that both factors, pre-acquired immunity and actual age of patients, favor the expression of CD36-binding variants. We have modified the paragraph accordingly and now say:

Line 202-211:

“Interestingly, however, in the same study CD36-binding variants are overrepresented in all groups of adult malaria patients analyzed, even in severe cases and in first-time infected individuals [84], which is in stark contrast to the pattern seen in severely ill children [70]. The authors speculate that this could be the reason for multisystemic disease symptoms in adult malaria patients. Alternatively, parasites in these less ill adult patients compared to children could have a less dominant expression of EPCR-binding PfEMP1 [84]. Further studies also showed that parasite cytoadhesion to CD36 correlates with the development of mild malaria [70,85,86]. Accordingly, both factors, the already acquired immunity and the age of patients seem to favor the expression of CD36-binding variants.”

  1. Line 204. The sentence currently reads “phenotypes that confer a developmental advantage for parasites”. I think this would be better described as a “selective” advantage, to avoid any confusion with the different developmental stages of the parasites.

Corrected – Line 216-218:

“Group A PfEMP1s are therefore thought to possess binding phenotypes that confer a selective advantage for parasites to replicate asexually, e.g., by decreasing splenic clearance, but at the same time favors the development of severe malaria [31].”

  1. The paragraph on lines 228-233 introduces the question of why P. falciparum has retained so many CD36 binding variants of PfEMP1. The authors might want to consider the analysis of Otto et al (Nature Microbiology, 2018) in which they showed that parasite species near the root of the Laverania (P. alderi, P. gaboni, P. blacklocki) do not possess CIDR-alpha domains within their PfEMP1s and therefore likely used different host receptors for cytoadherence. Only in the P. falciparum containing branch of the lineage is this domain (and presumably CD36 binding) expanded. This suggests that whatever selective advantage binding to CD36 confers, it arose at this particular point in evolution. Otto et al had no explanation for what this selective advantage was or why it arose when it did, and the authors of the submitted manuscript might not either. However, if the authors have any new thoughts or ideas on this puzzle, I would encourage them to include them.

This is a very interesting observation that we have now included in the text. Unfortunately, we also have no new ideas or could only speculate strongly about the selection advantage that has arisen.

Line 243-246:

“Interestingly, the CIDRα domains were shown to be present only in the P. falciparum-containing branch (clade B) of the Laverania subgenus. This could indicate that the binding to CD36 provides a selective advantage for P. falciparum [103]. What kind of selection advantage this was is yet unclear.”

  1. In Figure 3, are the colored balls below the endothelial layer meant to represent cytokines?

Line 365 (Figure 3):

The colored balls were labeled in the Figure 3 with cytokines/chemokines.

  1. In the paragraph on lines 357-267, the authors describe the importance of knobs for cytoadhesion. They might consider mentioning the work of Cholera et al (PNAS, 2008) who showed that hemoglobinopathies can alter knob morphology and thereby change PfEMP1 display and the strength of cytoadhesive properties. These results reinforce the importance of knobs.

We included the reference – Line 382-390.

“It is well established that knobs play a crucial role in the cytoadhesion of P. falciparum [45,47,48,127,135]. Among other findings, PfIEs from patients with hemoglobin S (HbS) or hemoglobin C (HbC; both hemoglobin mutations protect against severe complications and death from malaria [136-138] have been found to exhibit reduced cytoadhesion to microvascular endothelial cells [139,140]. The results suggest that HbS and HbC alter the erythrocyte membrane in a manner that inhibits the transport and/or docking of parasite proteins and impairs the ability of the parasite to remodel the surface of its host cell. This also leads to the fact that the knobs can no longer be formed correctly and PfEMP1 is also no longer presented correctly [139,140].”

  1. On line 379, the word knobs is repeated.

Line 392: Corrected

  1. Please check the legend to Figure 4. For example, there currently is no (D) and the other panel descriptions might be out of order.

Figure legend was corrected – Line 406-413:

Figure 4. Cytoadhesion of knobby and knobless PfIEs to transgenic CHO cells presenting CD36 on the cell surface (CHOCD36) under static and flow conditions and at different temperatures. (A) Adhesion of PfIEs (red; anti-glycophorin A) to CHOCD36 cells (blue: nucleus (DAPI), green: cell surface (CD36-GFP fusion protein) under static binding conditions. (B) Cytoadhesion of knobbless (-K) and knobby (+K) PfIEs to CHOCD36 cells . (C) Trajectories showing the rolling binding behavior of knobless (-K) and knobby (+K) PfIEs to CHOCD36 cells. (D) Average number of knobless (-K) and knobby (+K) PfIEs adhering to CHOCD36 cells at 37°C (blue) and 40°C (red) at a shear stress of 0.9 dyn/cm2 [127].

  1. Line 391. The sentence begins with “It”, which seems out of place.

Corrected.

Reviewer 2 Report

In the Review article Human CD36 and Plasmodium falciparum: The red queen hypothesis by Anna Bachmann, Nahla Galal Metwally, Johannes Allweier, Jakob Cronshagen, Maria del Pilar Martinez Tauler, Agnes Murk, Lisa Katharina Roth, Hanifeh Torabi1, Yifan Wu, Thomas Gutsmann and Iris Bruchhaus. Authors document that P. falciparum membrane proteins (PfEMP1) bind to endothelial receptors in the walls of human blood vessels. This binding prevents the parasite from reaching the spleen. In addition, the proteins bind to the CD36 receptor on various endothelial cells, arguing that the binding of PfIEs to CD36 regulates the balance of parasite growth and multiplication in most infections.

I have the following comments

On line 356 update the reference.

On line 104 please write P. falciparum in italics.

Please describe in more detail figure legend 1, it does not describe that it is shown in blue, green, yellow, red, etc.

Line 110 Define adequately what Knobs are, what they are made of and how they support the bond to the blood microvessels of different organs.

Lines 114 and 115 are redundant

The caption to figure 2 does not indicate how the microphotographs were obtained or what is shown.

In section 4 Endothelial cell receptors (ECRs); in addition to listing them, indicate which receptors are most important and why these receptors are important.

Discuss possible molecular interactions between CD36 binding and EPCRs that explain the differences in severe malaria complications.

It would help to include a figure on the involvement of CD36 in the immune response.

In line 379 the word knobs is repeated.

The conclusions describe the effects of internalization of CD36 bound to PfIEs by dendritic cells, such as activation of NK and T cells, but this information is absent in the text. Please do not conclude with new information not previously described.

Author Response

Response to Reviewer 2

We would like to thank the reviewer for the comments and suggestions for improvement. Below you will find our corresponding corrections.

On line 356 update the reference.

We have added references and described the part in more detail.

Line 373-381:

“It is well established that knobs play a crucial role in the cytoadhesion of P. falciparum [45,47,48,127,135]. Among other findings, PfIEs from patients with hemoglobin S (HbS) or hemoglobin C (HbC; both hemoglobin mutations protect against severe complications and death from malaria [136-138] have been found to exhibit reduced cytoadhesion to microvascular endothelial cells [139,140]. The results suggest that HbS and HbC alter the erythrocyte membrane in a manner that inhibits the transport and/or docking of parasite proteins and impairs the ability of the parasite to remodel the surface of its host cell. This also leads to the fact that the knobs can no longer be formed correctly and PfEMP1 is also no longer presented correctly [139,140].”

On line 104 please write P. falciparum in italics.

Corrected.

Please describe in more detail figure legend 1, it does not describe that it is shown in blue, green, yellow, red, etc.

Line 113-118:

The color code was included.

Figure 1. PfEMP1 repertoire of the P. falciparum isolate IT4, adapted from [17]. ECR binding phenotype [36]. Color code: Light brown: N-terminal segment (NTS); Bright Green: Duffy binding-like (DBL)α1; Light Blue: DBLα2, DBLα0; Dark Grey: Cys rich inter-domain regions (CIDR)α1; Light grey: CIDRα2-6; Dark green: CIDRγ; Orange: DBLβ; Yellow: DBLγ; Green: DBLδ; Pink: DBLε; Blue: DBLζ; Purple: DBLε; IT4var04: Light yellow: DBL/CIDRpam: Pink: DBLεpam.”

Line 110 Define adequately what Knobs are, what they are made of and how they support the bond to the blood microvessels of different organs.

In lines 119 to 131 (Section 3. Knobs – Anchor point for PfEMP1s), we describe exactly what knobs look like and which structural proteins they form. Section 13 (Importance of knobs for cytoadhesion) deals in detail with the importance of knobs for cytoadhesion (line 372-403).

Lines 114 and 115 are redundant

We have rephrased the sentences.

Line 124-127:

“Knobs are composed of various submembrane structural proteins, including the major protein of this structure, knob-associated histidine-rich protein (KAHRP). PfEMP3, ring-infected red cell antigen (RESA), mature parasite-infected red cell surface antigen (MESA)/PfEMP2, and Pf332 [41,42].”

The caption to figure 2 does not indicate how the microphotographs were obtained or what is shown.

We have now described the Figure in more detail (line 134-140):

Figure 2. The knobs of PflEs. (A) Transmission electron micrographs of uninfected and synchronised trophozoite-stage P. falciparum culture 24–28 h post-infection. The parasites were cultivated in the presence of human serum (10%) and the PfIEs were subjected to gelatin sedimentation to enrich knobby PfIEs [40,48]. (B) and (C) atomic force microscopic 3D-images of the surface of uninfected and trophozoite-stage PfIEs [40]. Images in (B) show magnifications directly from the membrane surface of the erythrocytes shown in (C). (D) Simplified schematic illustration of the structure of knobs.”

In section 4 Endothelial cell receptors (ECRs); in addition to listing them, indicate which receptors are most important and why these receptors are important.

In the section 4 (line 142-152), we listed the ECRs for which interaction with PfIEs is reported in the literature. For most of these receptors, there is no information whether and what role they play in malaria infection. Only in vitro binding of PfIEs to these ECRs has been described. Only the ECRs ICAM-1, EPCR and CD36 have been relatively well studied in the context of malaria. The knowledge is summarized in sections 6 (Pathology induced by cytoadhesion), 7 (ECR-specific expression in relation to the origin of the endothelial cells) and 8 (Hierarchy of var expression during the human blood phase).

Discuss possible molecular interactions between CD36 binding and EPCRs that explain the differences in severe malaria complications.

The most important difference, as also described in the manuscript, is based on the fact that endothelial cells originating from different organs present different ECRs on their cell surface. Thus, EPCRs are found on brain endothelial cells, whereas these do not show CD36. (see Section 7, ECR-specific expression in relation to the origin of the endothelial cells). Our working hypothesis is also that rolling over CD36 stimulates endothelial cells less (due to the short contact) than static binding to other ECRs. Since we are currently working on this question, statements about this would be pure speculation.

It would help to include a figure on the involvement of CD36 in the immune response.

Because the existing knowledge about the importance of CD36 in the context of immune defense is so far rather limited and because we have outlined the essential aspects in the text, we believe that a corresponding figure would not be helpful. This is mainly because it would include too many rather speculative aspects. These questions are currently engaging us in our research work, however, it is too early to discuss this.

In line 379 the word knobs is repeated.

Corrected.

The conclusions describe the effects of internalization of CD36 bound to PfIEs by dendritic cells, such as activation of NK and T cells, but this information is absent in the text. Please do not conclude with new information not previously described.

We have described the effect in detail under point F of section 11 (CD36 binding PfEMP1 variants - Benefits for parasite and host -).

F. DCs react to P. falciparum very early during infection and can thus influence the development of immunity. Internalization of PfIEs by DCs and subsequent pro-inflammatory cytokine production of DCs, NK and T cells depends on CD36. Notably, plasmacytoid DCs regulate innate and adaptive immunity to malaria via the production of proinflammatory cytokines. As this effect is particularly evident at low levels of parasitemia, the role of CD36 for malaria immunity appears to take place early during infection and promote the development of protective immunity against malaria [118,119].”

Reviewer 3 Report

In this review by Bachmann et al., the authors have described the host interacting receptors in Plasmodium falciparum, the parasite responsible for malaria. Based on the current title of the review, it is assumed that the authors wanted to present the host-parasite relationship in the light of the “Red Queen Hypothesis”. This hypothesis has been widely studied to describe the co-evolution of species in various contexts, including several pathogens and their interactions with host receptors. While the authors presented the existing knowledge about different host receptors including CD36 and the corresponding Plasmodium interacting proteins, the current version of the review falls short in explaining the coevolution of human CD36 protein and the parasite, as expected from the title. There are additional concerns with the current manuscript that need to be addressed.

1.     It has not been explained appropriately how the authors think the balance between the evolution of human CD36 and the interacting parasite proteins is maintained. To justify the use of the current title, this evolutionary relationship needs to be highlighted.

2.     Figure 2 – Indicate the scale in panel B. Are the two images representing infected erythrocytes from the same or different time points following infection? The figure legend for all the panels needs better explanation.

3.     Section 11- While the section highlighted the benefits of CD36 binding to the parasite, it does not explain the advantages experienced by the host. Consider explaining the parasite and host benefits of CD36 binding in separate sections.

4.     Figure 4 – Panel C – what do the different lines indicate? Consider showing a magnified image alongside the current image. Indicate Panel D in the legend.

5.     The authors should do a thorough review for the grammar, language check and typos throughout the manuscript. Scientific names should be italicized.

6.     Line 160 -164. The sentence is complex. Consider splitting the sentence.

7.     Line 197-199. The sentence is unclear. Consider splitting the sentence.

Author Response

Response to Reviewer 3

We would like to thank the reviewer for the comments and suggestions for improvement. Below you will find our corresponding corrections.

  1. It has not been explained appropriately how the authors think the balance between the evolution of human CD36 and the interacting parasite proteins is maintained. To justify the use of the current title, this evolutionary relationship needs to be highlighted.

After a very intensive discussion, we came to the conclusion that there is not enough solid data on the coevolution between CD36 and P. falciparum. Therefore, we decided to remove this section from the manuscript and to align the title exactly with the focus of the review.

New title: CD36 – a host receptor necessary for malaria parasites to establish and maintain infection

  1. Figure 2 – Indicate the scale in panel B. Are the two images representing infected erythrocytes from the same or different time points following infection? The figure legend for all the panels needs better explanation.

We have inserted a scale in Figure 2B. In addition, all figures have been described in more detail.

  1. Section 11- While the section highlighted the benefits of CD36 binding to the parasite, it does not explain the advantages experienced by the host. Consider explaining the parasite and host benefits of CD36 binding in separate sections.

We have thought about this for a long time, looking at the benefits for the parasite and the host separately, but have come to the conclusion that there are too many interconnections. We have only changed the order of the individual points, with the last three points being beneficial for both the parasite and the host:

  1. Cytoadhesion in non-vital tissues could promote survival and transmission of the parasite while minimising host damage and death.
  2. CD36-mediated binding of PfIEs increases the likelihood of phagocytosis by macrophages. This can lead to a reduction in parasitemia, but also allows the parasite to maintain a viable infection without causing too much damage to the host through high parasitemia.
  3. Plasmacytoid DCs regulate innate and adaptive immunity to malaria through the production of proinflammatory cytokines. As this effect is particularly evident at low levels of parasitemia, the effect of CD36 on malaria immunity appears to take place early during infection and promote the development of protective immunity against malaria.

  1. Figure 4 – Panel C – what do the different lines indicate? Consider showing a magnified image alongside the current image. Indicate Panel D in the legend.

We have explained Figure 4 in more detail and corrected the errors, Line 406-413:

Figure 4. Cytoadhesion of knobby and knobless PfIEs to transgenic CHO cells presenting CD36 on the cell surface (CHOCD36) under static and flow conditions and at different temperatures. (A) Adhesion of PfIEs (red; anti-glycophorin A) to CHOCD36 cells (blue: nucleus (DAPI), green: cell surface (CD36-GFP fusion protein) under static binding conditions. (B) Cytoadhesion of knobbless (-K) and knobby (+K) PfIEs to CHOCD36 cells . (C) Trajectories showing the rolling binding behavior of knobless (-K) and knobby (+K) PfIEs to CHOCD36 cells. (D) Average number of knobless (-K) and knobby (+K) PfIEs adhering to CHOCD36 cells at 37°C (blue) and 40°C (red) at a shear stress of 0.9 dyn/cm2 [127].”

  1. The authors should do a thorough review for the grammar, language check and typos throughout the manuscript. Scientific names should be italicized.

The complete manuscript was again revised in terms of grammar and linguistic style and improved accordingly.

  1. Line 160 -164. The sentence is complex. Consider splitting the sentence.

We have divided the sentence into three sentences.

Line 173-178:

“Examples include PfEMP1 variants that interact with ICAM-1 and EPCR or CD36. Dual binding to ICAM-1 and EPCR specifically enhances the binding of PfIEs to endothelial cells (ECs) under physiologically higher shear stresses. Expression of these variants has been associated with an increased risk of developing CM, including induction of brain swelling and disruption of the blood-brain barrier [36,74-76].”

  1. Line 197-199. The sentence is unclear. Consider splitting the sentence.

The sentence has been splitted.

Line 210-213:

“This broad expression pattern is modulated by existing host immunity. In African pre-exposed individuals, the expression of many variants at the parasite population level is reduced to very few or a single B-type, possibly reflecting gaps in the host antibody repertoire [91].”

Round 2

Reviewer 2 Report

The paper can be accepted

Reviewer 3 Report

In the revised submission, the authors have addressed all the concerns raised in the previous report and modified the manuscript appropriately. There is no further comment or concern with the manuscript.